# Impact of fitness coach behavior on exercise motivation, commitment, and enjoyment: A longitudinal study

Ricardo Braga-Pereira[1], Guilherme Eustáquio Furtado[2,3,4], Francisco Campos[2,4], António Rodrigues Sampaio[1] *, Pedro Teques[1,5]

1 N2i Research Centre, Polytechnic Institute of Maia, Maia, Portugal, 2 Polytechnic Institute of Coimbra, Coimbra, Portugal–Lagar dos Cortiços, Coimbra, Portugal, 3 Center for Studies on Natural Resources, Environment and Society (CERNAS), Polytechnic Institute of Coimbra, Coimbra, Portugal, 4 SPRINT—Sport Physical activity and health Research & INnovation cenTer, Polytechnic Institute of Coimbra, Coimbra, Portugal, 5 Research Center in Sports Sciences, Health Sciences and Human Development (CIDESD), Vila Real, Portugal

* arsampaio@umaia.pt

**Data Availability Statement:** All relevant data are within the manuscript.

**Funding:** The author G.E.F is funded by the National Foundation for Science and Technology

## Abstract

Fitness coaches seem to play an essential role in the field of exercise as they help prevent sedentary lifestyles and promote overall health, quality of life, and well-being. This study aimed to explore the effects of fitness coaches' behavior perceptions, intrinsic motivation, and enjoyment of exercisers on their long-term fitness and health, as well as their commitment to exercise. A total of 202 individuals participated in the study that was developed over three data gathering occasions (baseline, three months and six months). To this end, three psychometric scales were used as part of a multi-section survey: FCBS-Fit (perception of the fitness coach's behavior), IMIp (intrinsic motivation of exercisers), and PACES (enjoyment in exercise), along with the assessment of four health/fitness variables: anthropometry (i.e., waist-hip ratio), strength (i.e., handgrip strength), flexibility (i.e., sit-and-reach), and cardiorespiratory fitness (i.e., VO$_2$máx.). Overall, the results suggested that the perception of the fitness coach's behavior, the exerciser's intrinsic motivation, and enjoyment were predictors of the health/fitness outcomes, and seemed to be related to the exercise commitment at the end of three and six months. The novelty of this study is the examination of psychological and physiological factors in an integrated and longitudinal manner within the scope of exercise in fitness and health clubs. Therefore, these findings shed light on fitness coaches as an important element in the propagation and maintenance of exercise habits, accomplishment of tangible outcomes by exercisers, health promotion and the fitness sector's growth.

## Introduction

There is nowadays an agreement that engaging in regular and adequate physical activity and exercise across the lifetime can have a dramatic positive influence on health and well-being [1]. However, the population in general fails to reach the recommendations [2], which has become a significant public health issue [3]. Despite the initiatives conducted with the intent of

(FCT), P.I., through the institutional scientific employment program-contract (CEECINST/00077/2021). The funder had no role in study design, data collection and analysis, decision to publish, or preparation of the manuscript. This work was funded by National Funds by FCT - Foundation for Science and Technology under the following project UIDB/04045/2020 (https://doi.org/10.54499/UIDB/04045/2020).

**Competing interests:** The authors have declared that no competing interests exist.

enhancing exercise habits, the result ends in moderate and unsustained gains [4]. Hence, a more detailed understanding of the variables involved is needed to maximize the likelihood of successful involvement and adherence [5]. Once fitness coaches are at the forefront of health [6], the fitness sector and its body of professionals can become a game changer [7].

The enrollment and rates of abandonment at a fitness center are connected to improvements in fitness parameters. Failure to observe progress in less than a year leads to dropout [8]. It is expected that 40%-65% of individuals who start exercising dropout over the first 3–6 months [9–11] and only 30%-60% of members continue attending in the second year [12]. With withdrawing from exercise routines being one of the main challenges in the fitness sector [13], the quality of fitness coaches' behavior can be important in fostering better levels of intrinsic motivation in exercisers [14–16], enjoyment as one of the most essential concerns regarding exercise [17–20], supporting interpersonal behaviors [21,22], and satisfaction of basic psychological needs [23].

## Self-determination theory

This study uses Self-determination theory (SDT) as a conceptual framework [24–26]. SDT is a prevalent contemporary social-cognitive motivation theory in the fields of physical activity, sport, and exercise. According to this approach, social environments can either support or hinder human performance based on how well they meet individuals' basic psychological needs. SDT proposes that humans possess three fundamental psychological needs: autonomy (i.e., the ability to engage in behavior willingly), competence (i.e., the feeling of mastery and effectiveness), and relatedness (i.e., the desire for meaningful connections with others) [24,25]. Based on this notion, these basic needs are seen as crucial elements for people's adaptation, integrity, and development [27,28].

Moreover, the theory postulates that need-supportive social environments boost human beings' internal motivating sources and well-being. The combination of need-depriving (neglecting needs) and need-thwarting (actively undermining needs) social environments has a negative impact on the external sources of motivation in humans, leading to maladaptive outcomes such as ill-being [24,28]. Therefore, the social conditions mentioned above exist at contrasting extremes of a spectrum that establish the foundation for human motivation. Motivation to participate in an activity can be autonomous or controlled [25,26]. Autonomously motivation includes activities driven for inherent interest, enjoyment, and satisfaction (intrinsic motivation), or aligned with personal beliefs and valued outcomes (identified and integrated regulation, respectively).

Conversely, controlled motivation involves activities to avoid internal conflict (introjected regulation) or to react to external pressures (external regulation). Thus, motivation forms a continuum from lack of motivation to various levels of extrinsic and intrinsic motivation. Researchers have established that intrinsic motivation is the primary factor influencing long-term adherence to exercise [29]. Individuals with greater intrinsic motivation tend to exhibit more successful behaviors, such as increased adherence, and experience higher levels of health and psychological well-being [26]. Thus, this study examined the relationships between exercisers' intrinsic motivation, enjoyment, and the behavior of fitness coaches, considering that one of the key aspects of SDT is the evaluation of contextual factors [26].

In organized social environments (e.g., exercise), a social agent (e.g., fitness coach) can intentionally influence the social environment. As a result, the agent may support or influence the participants' motivation process through their behavior. In exercise settings, the coach's instruction style impacts participants' motivation [30]. The perception that the fitness coach has a supportive interaction style can favorably influence intrinsic motivation [31] and exercise

outcomes [32]. The behavior and emotional consequences of exercise can explain an individual's reasons for regularly engaging in exercise [33]. Individuals ought to perceive fitness coaches as supportive and exercising as an enjoyable activity to advocate long-term exercise commitment [34]. Hence, it is crucial to comprehend the factors involved to encourage regular and enduring engagement [35,36].

## Health and fitness

Achieving desired health benefits and improving physical fitness are the main drivers of long-term exercisers' use of health/fitness club [37]. Traditionally, health-related physical fitness involves four components: body composition, muscular fitness, flexibility, and cardiorespiratory fitness [38]. The waist-to-hip ratio [39] is a rapid and simple anthropometric assessment to identify an individual's body composition and degree of abdominal fat. Along with the Body Mass Index, it is a metric that quantifies obesity [40]. Nevertheless, the waist-to-hip ratio has been recommended as a superior predictor of cardiovascular disease [41].

Among the methods for testing muscle strength, handgrip strength has been widely used because it is a simple, rapid, cheap, and effective test that uses portable equipment [42]. Grip strength has been reported to be a consistent and valid predictor of health, muscular fitness, dexterity, and overall strength [43], among healthy and clinical populations [44]. This assessment could be used as a general indicator of overall muscle strength, including those of the lower limbs [45]. Flexibility pertains to the inherent characteristics of bodily tissues that establish maximal joint range of motion without resulting in injury [46]. Evaluating the flexibility of the hamstring muscles is crucial in the field of sports medicine because of its frequent correlation with prevalent non-contact injuries that can have both immediate and prolonged effects [47]. Hamstring flexibility should be incorporated into health-related testing protocols because of its significant relevance to both daily living activities and performance [38].

Cardiorespiratory fitness is a key predictor of mortality and morbidity [48]. It is considered one of the most significant measures of health, even above other traditional indicators such as body mass, blood pressure, and cholesterol levels [49]. Maximal oxygen consumption (VO$_2$máx.) is defined as the greatest rate at which oxygen can be transported, metabolized, and used during exercise. It is the best metric to define the overall ability of the cardiorespiratory system to supply oxygen during exercise [50]. Testing this physical capacity can help identify primary preventive needs and health promotion initiatives [51]. Given that the direct estimation of VO$_2$máx. has various restrictions (e.g., expensive and specialized equipment), this technique is impracticable in many cases. Field tests can be a viable option for estimating VO$_2$máx. with The Rockport Fitness Walking Test [52] being one of the submaximal assessments having the greatest validity [53].

To the best of our knowledge, no investigations have been undertaken utilizing this specific research design to examine the effects of fitness coaches on individuals' commitment to regular exercise. Therefore, the novelty of this study lies in its integrated and longitudinal analysis of both psychological and physiological aspects within the framework of exercise in health/fitness clubs. Thus, the purpose of this study is to analyze the influence of exercisers' perceptions of fitness coaches' behavior, intrinsic motivation, and enjoyment while exercising on over time fitness and health outcomes, specifically concerning anthropometry, strength, flexibility, and cardiorespiratory fitness. Additionally, the study aims to observe the relationship between these factors and adherence to exercise routines.

Based on the evidence provided [8,17,18,22,33,54], it is hypothesized that fitness coaches' behavior perceptions, intrinsic motivation, and enjoyment of exercisers might have the potential to influence health/fitness and, consequently, the sustainability of exercise routines.

## Materials and methods

### Participants

The research was conducted with Portuguese gym-goers engaged in fitness group classes (Aerobics, Cycling and Pilates), and/or in cardio-resistance training (which involved lifting weights or cardiovascular activities). Participants were selected on the basis of convenience from a pool of individuals judged relevant to the study's inquiry (i.e., their selection was driven by availability and their desire to participate). The study involved 202 participants recruited from September 2022 through June 2023, including both genders and varied age groups, all of whom actively volunteered to participate in the research. The dataset consists of 106 males (52.5%) and 96 females (47.5%), with ages spanning 18 to 64 years ($M_{age}$ = 36.17; SD = 12.30). On average, participants reported 7.45 years of exercise experience (SD = 8.93) and an average weekly attendance of 3.52 training sessions (SD = 1.35).

### Design and procedures

This research has a longitudinal quantitative design, considering three assessment stages (i.e., baseline (A1), three months (A2), and six months (A3)) to analyze the characteristics of the variables under study, identify their effects on the health/fitness results of the exercisers, and their commitment to exercise routines. The schedule of assessments is based on the significant known dropout rates that occur during these time periods [9–11].

Initially, potential exercisers in the fitness context were approached and requested to freely engage in the study (Fig 1). Of the 634 individuals who were first contacted, either directly or via email, 217 responded to the request. Throughout the recruitment process, 15 individuals were rejected because they displayed physical constraints that made it impossible for them to participate in the study or because they could not be scheduled. The remaining individuals were included after a sports anamnesis was performed. Subsequently, after streamlining the first proceedings, the assessments were arranged according to each participant's availability. In addition, the participants were requested to fill out a multi-section online survey.

A permanent contact channel was created between participants and researchers so that any questions could be answered swiftly. Participants were informed of the parameters and objectives of the study, their anonymity, and confidentiality. An informed and enlightened written consent to participate was provided. The participants completed the consent form expressing their agreement to participate in the study by answering affirmatively to the question: "I have read the information and give my authorization for my involvement in the study. Upon perusing the text, I comprehended the objectives of the investigation and was afforded the opportunity to clarify any doubts I may have had".

The study was approved by the Polytechnic Institute of Maia Research Center Scientific Committee (Ref. No. 003/06/22), and was conducted in accordance with the Declaration of Helsinki.

### Measures

To develop the design and implementation of the study, a multi-section survey was created to assess the independent variables. The instrument included three psychometric scales: perception of the fitness coach's behavior (i.e., FCBS-Fit), intrinsic motivation of the exerciser (i.e., IMIp), and enjoyment (i.e., PACES). Standardized and validated tests were employed to measure the dependent variables, which included: anthropometry (i.e., waist-hip ratio), strength (i.e., handgrip strength), flexibility (i.e., sit-and-reach), and cardiorespiratory fitness (i.e., $VO_2$máx.).

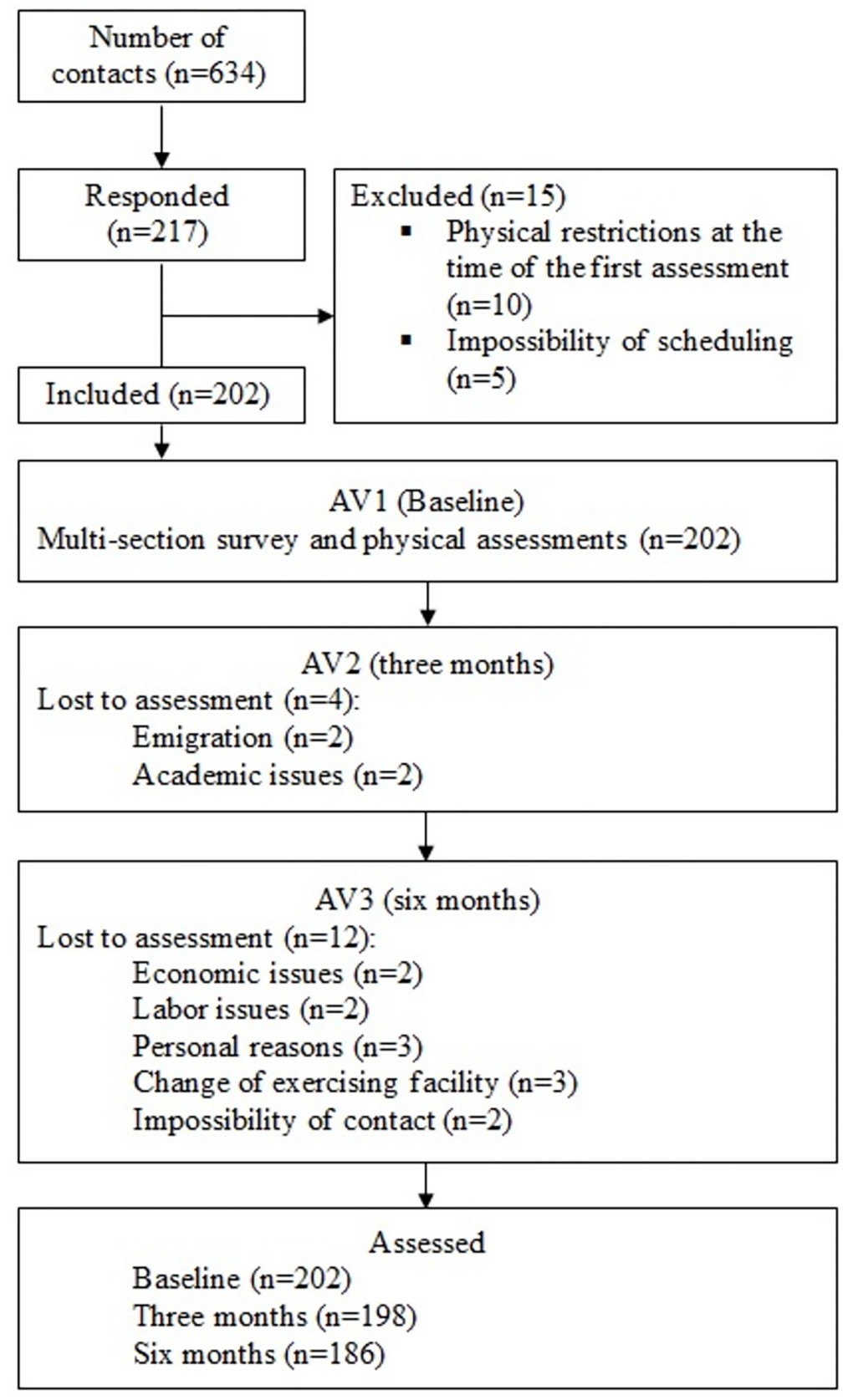

**Fig 1. Flowchart of the study participants.**

**Fitness coach behavior.** The Fitness Coaching Behavior Scale (FCBS-Fit) [55] aims to evaluate the quality of fitness coaches' behaviors by theoretically adopting the Coaching Model [56], which is commonly used in sports training. This measure comprises 22 items and is specifically designed for fitness coaches. The frequency of these events was measured using a 6-point Likert scale, including 1 (never does it), 2 (rarely), 3 (occasionally), 4 (many times), 5 (nearly usually), and 6 (always does it).

It is divided into four distinct dimensions: 1) Technical and positive rapport (providing demonstrations and instructions for exercises, offering corrective feedback, using positive reinforcement, encouraging continuous development, and fostering trust in the given instructions; 2) Goal setting (assist in establishing and defining both short- and long-term objectives and provide assistance in achieving them); 3) Exercise planning and prescription (involves creating customized, demanding, well-structured, and specific fitness programs for training sessions); and 4) Negative rapport (refers to behaviors such as yelling showing favoritism toward others, making ironic comments about one's inability, and making negative comments during exercise).

In general, the results from the different phases of psychometric analysis of FCBS-Fit confirm the validity and reliability of the data. In this study, the scale revealed the following internal consistency values: Cronbach's α was 0.82 for Technical and positive rapport; 0.88 for Goal setting; 0.77 for Exercise planning and prescription; and 0.73 for Negative rapport.

**Exerciser's intrinsic motivation.** Considering that intrinsic motivation is the primary factor influencing long-term exercise adherence [29], the Portuguese version of the Intrinsic Motivation Inventory (IMIp) [57] was used to evaluate the intensity of exercisers' intrinsic motivation in the context of physical sport and physical activity. The scale consists of 18 items grouped into four subscales: Pleasure (e.g., "I absolutely enjoy doing activities in the gym"; Competence (e.g., "I'm pleased with my performance in gym activities"); Effort (e.g., "I work hard in the gym"); and Pressure (e.g., "I feel under pressure while doing activities at the gym"). Participants expressed their motivation to practice one or more activities in the gym by agreeing or disagreeing with each of the statements and responded on a 5-point Likert scale (1 = Strongly disagree to 5 = Strongly agree).

This instrument has been frequently adopted in the realm of sports and exercise primarily because of its reliability and construct validity. In this study, the scale revealed the following internal consistency values: Cronbach's α was 0.79 for Enjoyment; 0.81 for Competence; 0.72 for Effort, and 0.88 for Pressure.

**Enjoyment.** For this purpose, the Portuguese version of the Physical Activity Enjoyment Scale (PACES) [58] was used. This measure consists of eight statements and assesses the amount of agreement with exercising enjoyment, based on a 7-point Likert scale ranging from "1-Strongly disagree" to "7-Strongly agree". This scale was originally developed to assess the enjoyment of physical activity using an 18-item scale [59]. More recently, Mullen and colleagues [60] validated an 8-item version with good psychometric properties from which the Portuguese validation was developed. In this study, the internal consistency determined by α was 0.87.

**Waist-hip ratio.** The ratio between waist and hip circumference is a superior clinical measure for predicting all-cause mortality from cardiovascular disease [61], and a reliable predictor of type II diabetes mellitus [62]. Previous research indicates that a waist-to-hip ratio greater than 0.90 cm for men and 0.85 cm for women is associated with an increased risk of diabetes mellitus and hypertension [63]. The buildup of excessive fat in the abdominal areas is related with a larger risk of diseases related to habits and body weight, while measuring the waist-to-hip ratio is an appropriate metric for this stratification [64].

Waist and hip circumferences were measured using a fabric tape measure, which normalizes the tension of the tape on the skin and increases the consistency of the measurement. The average of two measures was utilized as long as they did not vary by more than five millimeters; otherwise, the assessment would be repeated. As recommended, duplicate measurements at each site were obtained in rotation, rather than consecutively (i.e., taking a measurement at all the locations to be analyzed and then repeating the sequence) [38].

Waist circumference was determined by the participant standing, arms at their sides and along their body, feet together, and abdomen relaxed. A horizontal measurement was taken in the narrowest section of the torso (above the navel and below the xiphoid process). Concerning the hip circumference, with the individual standing and feet together, a horizontal measurement was obtained at the place of the greatest circumference of the glutes [38]. The ratio value of these two perimeters is the product of dividing the waist perimeter by the hip perimeter.

**Handgrip strength.** Handgrip strength is a measure of the functional condition of the upper extremity and is widely reported for a range of disorders [44]. In the clinical arena, it has been proven to be a significant instrument for detecting upper extremity impairments and a suitable guideline for treatment orientation [65,66]. It is favorably connected to bone mineral density [67] and has been suggested as a screening tool for osteoporosis concerns [68].

Longitudinal studies [69,70] reveal that low handgrip strength is a predictor of increased mortality from cardiovascular disease and cancer in males, even when characteristics related to muscle mass and body mass index are corrected. Handgrip strength is inversely associated with physical frailty, even when the effects of Body Mass Index and arm circumference are removed [71]. Researchers have hypothesized that factors connected to frailty and disability throughout life are related to muscular issues, which can be measured by handgrip dynamometry [71].

Measurements were conducted using a Jamar Hydraulic Hand Dynamometer (Sammons Preston Rolyan, Bolingbrook, IL, USA). Participants were asked to remain seated with their shoulders in a neutral position, one hand resting on the thigh and the elbow of the limb to be assessed flexed, forming a right angle, with the forearm in a neutral position [72] with the wrist between 0° and 30° of flexion and between 0° and 15° of ulnar deviation [73]. For all participants, the dynamometer handle was individually adjusted according to the size of their hands. The individuals were instructed to grasp the dynamometer with maximum effort in response to a standardized voice order and to press the handle for three seconds [74].

Measures for the dominant and nondominant limb were obtained in a randomized order. The rest period between each set was at least one minute and the highest value of the three attempts was selected for statistical analysis [75]. For the purposes of multiple linear regression, as the aim was to formalize a representative strength metric, the results of both limbs were averaged to calculate a total score.

**Flexibility.** The flexibility test was selected on the basis of its reliability and reproducibility [76,77]. The use of angle tests appears to be restricted in several situations because of the requirement for advanced apparatus, skilled specialists, and time limitations [78]. The assessment of fingertip distance in linear tests offers various benefits, such as it involves a straightforward technique, requires easily accessible materials, is simple to administer, and requires minimum application skills [79]. The sit-and-reach test is widely employed in physical fitness test batteries for measuring flexibility [80].

The examination was carried out using a box for the sit and reach test (40 cm×40 cm×34.5 cm). A scale was fastened to the top, perpendicular to the width of the box, for measurement. Participants sat on the floor with their legs extended at shoulder width and the balls of their feet against the box. With an overlapping hand, they flexed their hip joint and trunk to reach

as forward as possible, maintaining their knees, arms, and fingers completely extended. Participants completed three sets, resting for one minute between attempts, the best of which was utilized for statistical analysis. The recordings were made following a preliminary warm-up that comprised two initial practice attempts [38].

**Cardiorespiratory fitness.** The Rockport Fitness Walking Test [52] is one of the submaximal assessments with the greatest validity [53]. This test requires people to walk one mile (1600 m) as quickly as possible. After completing the test, the subject's age, height, body mass, terminal heart rate, and total trial length were entered into a regression equation developed to predict VO$_2$máx. [*VO$_2$máx. = 132.853 - (0.0769 × body mass)—(0.3877 × age) + (6.315 × gender)—(3.2649 × trial time)—(0.1565 × final heart rate)*]. The test has been validated for several populations, including healthy adults [81], women aged 65 and over [82], and high school students [83].

The test employed was the modified version suggested by Widrick and colleagues [84]. Employing this test on a treadmill enables a more practical and controlled evaluation, employing an ecological approach that is equally accurate as the field version. The strong correlation ($r = 0.91$) with a small total error of 5.26 ml/kg/min of oxygen indicates that the predictive equation previously developed for the field can be equally successful when applied to the modified version conducted on the treadmill. The ergometer utilized in this investigation was a FFITTECH RUN-T100 (Taipei, TW, China).

The protocol entailed a short duration of initial adjustment and speed determination, lasting approximately five minutes. The participants were directed to choose a rapid tempo that they could sustain for 15–20 minutes. Thus, the test could be conducted while maintaining a constant pace. Continuous monitoring of heart rate was conducted using a Polar H7 Wireless Heart Rate Sensor Band (Bethpage, NY, USA). The final trial time and heart rate were recorded once the specified distance was reached. One attempt is deemed adequate because the walking test results demonstrate high test-retest reliability, yielding findings almost similar to those obtained from a single test [52].

## Statistical analysis

Previous to data collection, an a priori statistical power analysis was performed to determine the required sample size. The primary objective of this analysis was to ensure that the research had sufficient statistical power to identify significant effects, if they occurred within the variables being investigated, while considering the specific features of this research. The software employed for this purpose was G*Power version 3.1.9.7 [85].

The analysis incorporated a medium effect size ($d = 0.5$) to depict the difference between two dependent means and multiple linear regression ($f^2 = 0.15$). The employed values were determined considering previous findings, which indicated effect magnitudes ranging from small to medium magnitudes [33,86]. The significance level ($\alpha$) was established at 0.05, acting as the cutoff point for rejecting the null hypothesis. The statistical power ($1–\beta$) was calculated at 0.80, suggesting an 80% likelihood of uncovering a real effect. Two-tailed tests were utilized for the analysis. The paired sample *t*-test suggested a sample size of 34 participants; however, multiple linear regression analysis with four predictors showed that 85 individuals were needed.

Once the data had been collected, it was processed and analyzed using the IBM SPSS Statistics software. For analyzing the descriptive statistics, data were acquired on sociodemographic factors (mean, minimum, maximum and standard deviation). A paired sample *t*-test was employed to analyze the research variables' dynamics over time. For the predictive analysis, multiple linear regression was performed to evaluate whether independent variables could

predict the dependent variables (health/fitness) investigation and what the impact would be in terms of exercise routines maintenance.

## Results

Table 1 provides general data on the descriptive statistics of the variables under study at the three evaluation moments.

To examine the characteristics of the variables over time (after three months and after six months), an analysis was carried out using the paired sample $t$-test, which showed that the perception of the fitness coach' behavior (e.g., goal setting; $[t(198) = -5.45; p < 0.01; d = 0.39]$–A2; $[t(186) = -5.76; p < 0.01; d = 0.44]$–A3), the exerciser's intrinsic motivation (e.g., competence; $[t(198) = -6.34; p < 0.01; d = 0.44]$–A2; $[t(186) = -3.30; p < 0.01; d = 0.37]$), and enjoyment $[t(198) = -5.14; p < 0.01; d = 0.37]$–A2; $[t(198) = -3.78; p < 0.01; d = 0.28]$) seem to be related to exercise commitment in both time frames (Tables 2 and 3).

Subsequently, a multiple linear regression analysis was conducted to examine whether the perception of the fitness coach's behavior and the exerciser's intrinsic motivation and enjoyment were able to predict the health/fitness outcomes (i.e. anthropometry, strength, flexibility, and cardiorespiratory fitness). As demonstrated in Tables 4–7, the analysis resulted in statistically significant models for the variables at three and six months.

Considering the period of three months (A2), the categories "pressure" of the exerciser's intrinsic motivation ($\beta = 0.18; t = 2.52; p < 0.01$) and "enjoyment" related to the exerciser's enjoyment of the exercise ($\beta = 0.29; t = 4.30; p < 0.01$) are the best predictors of the results obtained in the independent variable related to anthropometry (i.e., waist-to-hip ratio). The "competence" dimension of the exerciser's intrinsic motivation ($\beta = 0.24; t = 2.88; p < 0.01$) was the strongest predictor of the results obtained in the handgrip strength test.

The categories "goal setting" regarding the perception of the fitness coach's behavior ($\beta = 0.34; t = 3.05; p < 0.01$), "competence" regarding the exerciser's intrinsic motivation ($\beta = 0.17;$

**Table 1. Statistics of the different variables (M ± SD).**

| Variables | | A1 | | A2 | | A3 | |
|---|---|---|---|---|---|---|---|
| | | n | M ± SD | n | M ± SD | n | M ± SD |
| FCBS-FIT | TPR | 202 | 4.35±1.70 | 198 | 4.59±1.48 | 186 | 4.82±1.18 |
| | GS | | 3.79±1.81 | | 4.12±1.60 | | 4.54±1.27 |
| | EPP | | 4.31±1.79 | | 4.52±1.54 | | 4.74±1.27 |
| | NR | | 1.32±0.83 | | 1.25±0.64 | | 1.22±0.58 |
| IMIp | P | 202 | 4.15±0.91 | 198 | 4.25±0.83 | 186 | 4.25±0.83 |
| | C | | 3.13±0.68 | | 3.45±0.84 | | 3.46±0.86 |
| | E | | 3.05±0.59 | | 3.27±0.69 | | 3.27±0.71 |
| | P | | 2.44±0.70 | | 2.21±0.83 | | 2.20±0.84 |
| PACES | EJ | 202 | 4.70±0.82 | 198 | 4.91±0.86 | 186 | 4.94±0.87 |
| ANTH | WHR | 202 | 0.82±0.08 | 198 | 0.81±0.08 | 186 | 0.81±0.08 |
| STR | HS | 202 | 37.21±11.86 | 198 | 38.41±11.58 | 186 | 39.71±10.72 |
| FLX | SR | 202 | 18.62±4.62 | 198 | 20.19±4.40 | 186 | 20.37±4.16 |
| CFIT | VM | 202 | 37.77±9.89 | 198 | 39.95±9.26 | 186 | 41.27±8.66 |

Note: n = sample size; M = mean; SD = standard deviation; TPR = technical and positive rapport; GS = goal setting; EPP = exercise planning and prescription; NR = negative rapport; P = pleasure; C = competence; E = effort; P = pressure; EJ = enjoyment; WHR = waist-hip ratio; HS = handgrip strength; SR = sit-and-reach; VM = VO$_2$máx.; ANTH = anthropometry; STR = strength; FLX = flexibility; CFIT = cardiorespiratory fitness; A1 = assessment 1 (baseline); A2 = assessment 2 (three months); A3 = assessment 3 (six months).

**Table 2. Paired sample *t*-test data (three months).**

| Variables | | M ± SD | CI (95%) | | *t* | Δ% | *P* | *d* |
|---|---|---|---|---|---|---|---|---|
| | | | Inf. | Sup. | | | | |
| FCBS-FIT | TPR | -0.25±0.79 | -0.36 | -0.14 | -4.37 | 5.52 | 0.00** | 0.32 |
| | GS | -0.34±0.88 | -0.47 | -0.22 | -5.45 | 8.71 | 0.00** | 0.39 |
| | EPP | -0.22±0.80 | -0.33 | -0.11 | -3.83 | 4.87 | 0.00** | 0.28 |
| | NR | 0.08±0.45 | 0.01 | 0.14 | 2.37 | 5.30 | 0.02* | 0.18 |
| IMIp | P | -0.11±0.57 | -0.19 | -0.03 | -2.60 | 2.41 | 0.01** | 0.19 |
| | C | -0.32±0.72 | -0.42 | -0.22 | -6.34 | 10.22 | 0.00** | 0.44 |
| | E | -0.21±0.60 | -0.29 | -0.13 | -5.04 | 7.21 | 0.00** | 0.35 |
| | P | 0.23±0.61 | 0.14 | 0.31 | 5.26 | 9.43 | 0.00** | 0.38 |
| PACES | EJ | -0.22±0.59 | -0.30 | -0.13 | -5.14 | 4.47 | 0.00** | 0.37 |
| ANTH | WHR | 0.01±0.04 | 0.01 | 0.02 | 3.56 | 1.22 | 0.00** | 0.25 |
| STR | HS R | -1.60±3.35 | -2.07 | -1.13 | -6.73 | 3.23 | 0.00** | 0.48 |
| | HS L | -0.79±2.46 | -1.14 | -0.45 | -4.55 | 8.43 | 0.00** | 0.32 |
| FLX | SR | -1.55±2.99 | -1.97 | -1.13 | -7.29 | 5.77 | 0.00** | 0.52 |
| CFIT | VM | -2.15±4.60 | -2.79 | -1.51 | -6.60 | 5.52 | 0.00** | 0.47 |

Note

*. *p* < 0.05

**. *p* < 0.01; M = mean; SD = standard deviation; CI = confidence interval; Inf = inferior; Sup = superior; *t* = t-value; Δ% = percent variance; *p* = p-value; *d* = effect size; TPR = technical and positive rapport; GS = goal setting; EPP = exercise planning and prescription; NR = negative rapport; P = pleasure; C = competence; E = effort; P = pressure; EJ = enjoyment; WHR = waist-hip ratio; HS R = handgrip strength (right limb); HS L = handgrip strength (left limb); SR = sit-and-reach; VM = VO$_2$máx.; ANTH = anthropometry; STR = strength; FLX = flexibility; CFIT = cardiorespiratory fitness.

**Table 3. Paired sample *t*-test data (six months).**

| Variables | | M ± SD | CI (95%) | | *t* | Δ% | *P* | *d* |
|---|---|---|---|---|---|---|---|---|
| | | | Inf. | Sup. | | | | |
| FCBS-FIT | TPR | -0.21±0.90 | -0.34 | -0.08 | -3.19 | 5.01 | 0.00** | 0.23 |
| | GS | -0.43±1.02 | -0.57 | -0.28 | -5.76 | 10.19 | 0.00** | 0.44 |
| | EPP | -0.18±0.75 | -0.29 | -0.07 | -3.33 | 4.87 | 0.00** | 0.24 |
| | NR | 0.03±0.37 | -0.03 | 0.08 | 1.00 | 2.40 | 0.32 | |
| IMIp | P | -0.10±0.42 | -0.16 | -0.04 | -3.30 | 0.25 | 0.00** | 0.24 |
| | C | -0.19±0.51 | -0.16 | -0.04 | -3.30 | 0.29 | 0.00** | 0.37 |
| | E | -0.08±0.33 | -0.13 | -0.03 | -3.36 | 0.20 | 0.00** | 0.24 |
| | P | 0.20±0.54 | 0.12 | 0.28 | 5.04 | 0.45 | 0.00** | 0.37 |
| PACES | EJ | -0.11±0.39 | -0.16 | -0.05 | -3.78 | 0.61 | 0.00** | 0.28 |
| ANTH | WHR | 0.00±0.05 | 0.01 | -0.01 | 0.01 | 0 | 0.60 | |
| STR | HS R | -1.01±3.16 | -1.46 | -0.55 | -4.35 | 3.39 | 0.00** | 0.32 |
| | HS L | -1.01±3.08 | -1.45 | -0.56 | -4.46 | 0.89 | 0.00** | 0.33 |
| FLX | SR | -0.10±1.14 | -0.26 | 0.07 | -1.15 | 3.30 | 0.25 | |
| CFIT | VM | -1.31±3.59 | -1.82 | -0.79 | -4.97 | 5.01 | 0.00** | 0.36 |

Note

*. *p* < 0.05

**. *p* < 0.01; M = mean; SD = standard deviation; CI = confidence interval; Inf = inferior; Sup = superior; *t* = t-value; Δ% = percent variance; *p* = p-value; *d* = effect size; TPR = technical and positive rapport; GS = goal setting; EPP = exercise planning and prescription; NR = negative rapport; P = pleasure; C = competence; E = effort; P = pressure; EJ = enjoyment; WHR = waist-hip ratio; HS R = handgrip strength (right limb); HS L = handgrip strength (left limb); SR = sit-and-reach; VM = VO$_2$máx.; ANTH = anthropometry; STR = strength; FLX = flexibility; CFIT = cardiorespiratory fitness.

**Table 4. Multiple linear regression data (anthropometry).**

| Variables | | Three months | | | | | | Six months | | | | | |
|---|---|---|---|---|---|---|---|---|---|---|---|---|---|
| | | $R^2$ | $F$ | $p$ | $\beta$ | $t$ | $p$ | $R^2$ | $F$ | $P$ | $\beta$ | $t$ | $p$ |
| FCBS-FIT | TPR | 0,05 | 2,48 | 0,05* | -0,09 | -0,78 | 0,44 | 0,04 | 1.83 | 0.13 | -0,18 | -1,72 | 0,09 |
| | GS | | | | -0,14 | -1,20 | 0,23 | | | | 0,05 | 0,50 | 0,62 |
| | EPP | | | | 0,01 | 0,07 | 0,94 | | | | -0,05 | -0,45 | 0,65 |
| | NR | | | | 0,07 | 0,91 | 0,36 | | | | 0,04 | 0,51 | 0,61 |
| IMIp | P | 0,08 | 4,22 | 0,00** | 0,04 | 0,50 | 0,62 | 0.02 | 0.93 | 0.45 | -0,03 | -0,36 | 0,72 |
| | C | | | | -0,14 | -1,76 | 0,08 | | | | -0,07 | -0,91 | 0,37 |
| | E | | | | -0,14 | -1,73 | 0,09 | | | | -0,08 | -0,98 | 0,33 |
| | P | | | | 0,18 | 2,52 | 0,01** | | | | 0,05 | 0,60 | 0,55 |
| PACES | EJ | 0,09 | 18,49 | 0,00** | -0,29 | -4,30 | 0,00** | 0.02 | 5.06 | 0.03* | -0,16 | -2,25 | 0,03* |

Note

*. $p < 0.05$

**. $p < 0.01$; $R^2$ = explained variance; $F$ = F-value; $\beta$ = standardized beta coefficient; $t$ = t-value; $p$ = p-value; TPR = technical and positive rapport; GS = goal setting; EPP = exercise planning and prescription; NR = negative rapport; P = pleasure; C = competence; E = effort; P = pressure; EJ = enjoyment.

Three months: [$F_{(4,194)}$ = 2.48; $p < 0.05$; $R^2$ = 0.05]–Fitness coach behavior; [$F_{(4,194)}$ = 4.22; $p < 0.01$; $R^2$ = 0.08]–Exerciser intrinsic motivation; [$F_{(1,197)}$ = 18.49; $p < 0.01$; $R^2$ = 0.09]–Enjoyment.

Six months: [$F_{(1,185)}$ = 5.06; $p < 0.05$; $R^2$ = 0.02]–Enjoyment.

$t$ = 2.11; $p < 0.05$), and "enjoyment" ($\beta$ = 0.26; $t$ = 3.72; $p < 0.01$) were the best predictors of the results obtained in the flexibility test (sit-and-reach). The categories "goal setting" ($\beta$ = 0.31; $t$ = 2.71; $p < 0.01$) and "exercise planning and prescription" ($\beta$ = 0.21; $t$ = 1.96; $p < 0.05$), both linked to the perception of the fitness coach's behavior, are the best predictors of the results obtained in cardiorespiratory fitness (VO$_2$máx.).

With regard to the six-month period (A3), the "enjoyment" dimension connected to the exerciser's enjoyment in exercising ($\beta$ = 0.16; $t$ = 2.25; $p < 0.05$) is the one that best predicts

**Table 5. Multiple linear regression data (strength).**

| Variables | | Three months | | | | | | Six months | | | | | |
|---|---|---|---|---|---|---|---|---|---|---|---|---|---|
| | | $R^2$ | $F$ | $p$ | $\beta$ | $t$ | $p$ | $R^2$ | $F$ | $P$ | $\beta$ | $t$ | $p$ |
| FCBS-FIT | TPR | 0,00 | 0,13 | 0,97 | 0,03 | 0,23 | 0,82 | 0,04 | 4,37 | 0,00** | -0,09 | -0,82 | 0,02* |
| | GS | | | | 0,03 | 0,26 | 0,79 | | | | 0,00 | 0,02 | 0,98 |
| | EPP | | | | -0,00 | -0,02 | 0,99 | | | | 0,14 | 1,33 | 0,03* |
| | NR | | | | 0,01 | 0,13 | 0,90 | | | | 0,03 | 0,36 | 0,72 |
| IMIp | P | 0,06 | 3,10 | 0,02* | -0,01 | -0,08 | 0,93 | 0,03 | 1,18 | 0,32 | 0,02 | 0,32 | 0,75 |
| | C | | | | 0,24 | 2,88 | 0,00** | | | | 0,11 | 1,43 | 0,16 |
| | E | | | | 0,03 | 0,34 | 0,73 | | | | 0,08 | 1,01 | 0,31 |
| | P | | | | 0,01 | 0,09 | 0,93 | | | | 0,01 | 0,09 | 0,93 |
| PACES | EJ | 0,00 | 0,45 | 0,50 | 0,05 | 0,67 | 0,50 | 0,00 | 0,00 | 0,99 | 0,00 | 0,01 | 0,99 |

Note

*. $p < 0.05$

**. $p < 0.01$; $R^2$ = explained variance; $F$ = F-value; $\beta$ = standardized beta coefficient; $t$ = t-value; $p$ = p-value; TPR = technical and positive rapport; GS = goal setting; EPP = exercise planning and prescription; NR = negative rapport; P = pleasure; C = competence; E = effort; P = pressure; EJ = enjoyment.

Three months: [$F_{(4,194)}$ = 3.10; $p < 0.05$; $R^2$ = 0.06]–Exerciser intrinsic motivation.

Six months: [$F_{(4,182)}$ = 4.37; $p < 0.01$; $R^2$ = 0.04]–Fitness coach behavior.

**Table 6. Multiple linear regression data (flexibility).**

| Variables | | Three months | | | | | | Six months | | | | | |
|---|---|---|---|---|---|---|---|---|---|---|---|---|---|
| | | $R^2$ | F | p | β | t | p | $R^2$ | F | P | β | t | p |
| FCBS-FIT | TPR | 0,06 | 3,14 | 0,02* | -0,13 | -1,16 | 0,25 | 0,02 | 1,12 | 0,35 | -0,03 | -0,32 | 0,75 |
| | GS | | | | 0,34 | 3,05 | 0,00** | | | | 0,17 | 1,67 | 0,10 |
| | EPP | | | | -0,03 | -0,33 | 0,74 | | | | -0,11 | -1,01 | 0,32 |
| | NR | | | | 0,04 | 0,53 | 0,60 | | | | 0,08 | 1,07 | 0,29 |
| IMIp | P | 0,05 | 2,67 | 0,03* | 0,00 | 0,01 | 0,99 | 0,02 | 0,93 | 0,45 | 0,05 | 0,63 | 0,53 |
| | C | | | | 0,17 | 2,11 | 0,04* | | | | 0,03 | 0,43 | 0,67 |
| | E | | | | 0,06 | 0,73 | 0,47 | | | | 0,03 | 0,37 | 0,71 |
| | P | | | | -0,09 | -1,25 | 0,21 | | | | -0,10 | -1,40 | 0,16 |
| PACES | EJ | 0,07 | 13,80 | 0,00** | 0,26 | 3,72 | 0,00** | 0,02 | 3,95 | 0,05* | 0,15 | 1,99 | 0,05* |

Note

*. $p < 0.05$

**. $p < 0.01$; $R^2$ = explained variance; $F$ = F-value; β = standardized beta coefficient; $t$ = t-value; $p$ = p-value; TPR = technical and positive rapport; GS = goal setting; EPP = exercise planning and prescription; NR = negative rapport; P = pleasure; C = competence; E = effort; P = pressure; EJ = enjoyment.

Three months: $[F(4,194) = 3,14$; $p < 0,05$; $R^2 = 0,06]$–Fitness coach behavior; $[F(4,194) = 2,67$; $p < 0,05$; $R^2 = 0,05]$–Exerciser intrinsic motivation; $[F(1,197) = 13,80$; $p < 0,01$; $R^2 = 0,07]$–Enjoyment.

Six months: $[F(1,185) = 3,95$; $p < 0,05$; $R^2 = 0,02]$–Enjoyment.

the results obtained in the waist-hip ratio. With relation to handgrip strength, the dimensions "technical and positive rapport" (β = 0.09; $t$ = 0.82; $p < 0.05$) and "exercise planning and prescription" (β = 0.14; $t$ = 1.33; $p < 0.05$), on the perception of the fitness coach's behaviors, were the greatest predictors of the results observed.

The "enjoyment" (β = 0.15; $t$ = 1.99; $p < 0.05$) is the strongest predictor of the results achieved in the sit-and-reach test. Finally, the categories "technical and positive rapport" (β = 0.06; $t$ = 0.59; $p < 0.05$) and "exercise planning and prescription" (β = 0.04; $t$ = 0.38; $p < 0.01$), both relating to the perception of the fitness coach's behavior, and the "competence"

**Table 7. Multiple linear regression data (cardiorespiratory fitness).**

| Variables | | Three months | | | | | | Six months | | | | | |
|---|---|---|---|---|---|---|---|---|---|---|---|---|---|
| | | $R^2$ | F | p | β | t | p | $R^2$ | F | P | β | t | p |
| FCBS-FIT | TPR | 0,06 | 2,81 | 0,03* | -0,01 | -0,12 | 0,91 | 0,05 | 3,27 | 0,00** | -0,06 | -0,59 | 0,05* |
| | GS | | | | 0,31 | 2,71 | 0,01** | | | | 0,17 | 1,60 | 0,11 |
| | EPP | | | | -0,21 | -1,96 | 0,05* | | | | -0,04 | -0,38 | 0,01** |
| | NR | | | | -0,11 | -1,53 | 0,13 | | | | -0,12 | -1,64 | 0,10 |
| IMIp | P | 0,02 | 1,20 | 0,31 | -0,05 | -0,65 | 0,52 | 0,06 | 4,53 | 0,03* | -0,01 | -0,18 | 0,86 |
| | C | | | | 0,09 | 1,10 | 0,27 | | | | 0,04 | 0,56 | 0,03* |
| | E | | | | 0,10 | 1,24 | 0,22 | | | | 0,06 | 0,77 | 0,44 |
| | P | | | | -0,06 | -0,82 | 0,41 | | | | 0,01 | 0,13 | 0,90 |
| PACES | EJ | 0,01 | 1,45 | 0,23 | 0,31 | 2,71 | 0,23 | 0,01 | 1,61 | 0,21 | 0,09 | 1,27 | 0,21 |

Note

*. $p < 0.05$

**. $p < 0.01$; $R^2$ = explained variance; $F$ = F-value; β = standardized beta coefficient; $t$ = t-value; $p$ = p-value; TPR = technical and positive rapport; GS = goal setting; EPP = exercise planning and prescription; NR = negative rapport; P = pleasure; C = competence; E = effort; P = pressure; EJ = enjoyment.

Three months: $[F(4,194) = 2,81$; $p < 0,05$; $R^2 = 0,06]$–Fitness coach behavior.

Six months: $[F(4,182) = 3,27$; $p < 0,01$; $R^2 = 0,05]$–Fitness coach behavior; $[F(4,182) = 4,53$; $p < 0,05$; $R^2 = 0,06]$–Exerciser intrinsic motivation.

dimension of the exerciser's intrinsic motivation ($\beta = 0.04$; $t = 0.56$; $p < 0.05$), are the best predictors of the results obtained in cardiorespiratory fitness (VO$_2$máx.).

## Discussion

This manuscript aimed to investigate the impact of exercisers' perceptions on fitness coaches' behavior, intrinsic motivation, and enjoyment during exercise on over time health and fitness outcomes (i.e., anthropometry; strength; flexibility; cardiorespiratory fitness). Furthermore, this study seeks to examine the relationship between these factors and the long-lasting nature of exercise routines. The primary hypothesis was confirmed by the main outcomes, which revealed that exercisers' perceptions of the fitness coach's behavior, intrinsic motivation, and enjoyment during exercise seem to have an important influence on health/fitness, and consequently the continual upkeep of exercise routines.

After the first three-month period (AV2), both independent variables (perception of the fitness coach's behavior, the exerciser's intrinsic motivation and enjoyment) showed similar prediction ability for the dependent variables (i.e. waist-hip ratio, handgrip strength, flexibility and cardiorespiratory fitness). Three of the dimensions were related to the fitness coach's behavior ("goal setting" twice, and "exercise planning and prescription"), three were related to the exerciser's intrinsic motivation ("competence" twice, and "pressure"), and two were related to the exerciser's enjoyment.

However, after six months (AV3), the perception of the fitness coach's behavior emerged more prominently. Overall, four categories emerged ("technical and positive rapport" and "exercise planning and prescription" twice). The exerciser's intrinsic motivation was represented once by the dimension "competence". The exerciser's enjoyment was represented twice. It should be highlighted that over time, the perception of the fitness coach's behaviors were the variable with the highest power to predict the outcomes for exercisers. "Technical and positive rapport" emerges more prominently in the analysis over time, whereas "goal setting" were of greater importance in the initial period (three months).

Overall, competence, enjoyment, and exercise planning and prescription are the categories with the greatest predictive capacity for exercisers' outcomes. Competence (i.e. a person's ability to be efficient in interacting with the environment) is one of the essential nutrients of basic psychological needs that is important in the development of intrinsic motivation [25,26] and consequently determines how a person regulates their behavior when engaging in structured physical activities [15,87].

Satisfying this need, which is crucial for psychological health, by engaging in physical activities has a direct influence on the enjoyment of those who exercise [26]. Being innate in all human beings, an individual will attempt to engage in activities that produce experiences and sensations of competence. This underlying desire is called intrinsic motivation. An individual whose basic psychological needs are addressed will be naturally motivated toward that activity [29].

Enjoyment is positively associated with several psychological and behavioral factors among exercisers, including intrinsic motivation [58,88]. It has been claimed that enjoyment is a vital factor for understanding and clarifying motivation and individuals' experiences in the realm of exercise [5,89,90]. It is distinguished by a positive attitude towards exercise being one of the most essential factors [20], a predictor and outcome [20,91].

Also, is substantially connected with fitness coach skills [92]. Exercisers' perceptions of the coaches' interaction style alter their intrinsic motivation, depending on their perceived competence and motivation [93]. The feeling that the fitness coach has a supportive interaction style can favorably improve intrinsic motivation to exercise [22]. Individuals who are more

intrinsically motivated tend to enjoy things more, have greater well-being, and exhibit better adaptive behavior. In this approach, intrinsic motivation can impact exercise consistency [14,15], and the ensuing outcomes [94].

Of the total number of participants who started the study, only 16 were lost throughout the process. These results seem to contrast with the high rates reported in the literature within three and six months [9–11]. Thus, the findings of this study may strengthen the academic discussion regarding the potential impact of the fitness coach's behavior on individuals' commitment to exercise regimens, encompassing their intrinsic motivation, enjoyment, and enhancement of their health and fitness parameters.

## Study limitations and recommendations for future research

The findings arising from this research can contribute valuable insights to the field by delving into how specific dimensions of fitness coach behavior, exerciser intrinsic motivation, and enjoyment are associated with health/fitness outcomes, thus enriching the understanding of this critical aspect of exercise commitment within the realm of fitness and health club day-by-day dynamics. Nevertheless, this study has limitations. First, this study did not control for how long the exercisers had been active in health/fitness clubs exercise routines at the outset of the study (i.e. whether they were novices or already got involved in exercising). This may have interfered with the exercisers' perceptions, given the length of time they have been in contact with the fitness coaches and the fact that individuals who have been dealing with them for longer may be more acquainted with them.

In addition, concerns connected to the exerciser's own intrinsic motivation and enjoyment, whether they are more or less experienced, or just beginning out. With relation to the concept of dropout, it would also be crucial to enroll only individuals who were starting to exercise at the exact moment that the study began. However, the insertion of inclusion and/or exclusion criteria on what is characterized as the limiting features would impact the number of participants available. In the future, it may be interesting to recruit participants from different locations, milieus, and circumstances (e.g., studios, *boxes*, at-home exercise, and outdoor exercise). This will make it possible to broaden geographic coverage and enable a more comprehensive knowledge of the effects of fitness coach behavior, intrinsic motivation, and enjoyment in many different settings.

Regarding the scope of the study, raising it to one year may be meaningful, as coupled with the time frame of three and six months, it is regarded as a critical period in the literature [11,12]. Likewise, with a longer length, new data may emerge, which would allow a more exact identification of changes in health/fitness outcomes over time, helping to determine the causal relationship between the variables evaluated and enrollment in exercising programs. Additionally, future research should explore experimental study designs aiming to examine the effects of coaching strategies and motivation- and/or enjoyment-building programs in health/fitness outcomes, and consequently in exercise maintenance. With reference to health/fitness variables, other indicators such as the rate-pressure product (i.e., product of heart rate and systolic blood pressure), which is a commonly used trustworthy clinical mark of myocardial oxygen demand, could be explored.

## Practical applications

This study highlights the importance of fitness coaches adopting supportive interpersonal behaviors and a motivational climate because its role as a predictor of adherence to exercise is well known [5,18,95], so that exercisers can acquire more self-determined (intrinsic) forms of motivation to provide them with a better experience that fosters long-term commitment

[29,33,93]. By the other hand, a dynamic link between exercise and health-related parameters is known [96]. One of these elements is physical fitness, which refers to the ability to perform exercises. Physical fitness is a multidimensional notion and incorporates health-related components such as cardiorespiratory fitness, strength, and flexibility [97]. Current research [98,99] has indicated positive relationships between physical fitness and several health-related outcomes.

Some factors connected to health and fitness are related to the risk of withdrawing from exercise [11,100]. In this sense, this study presents some information that could be beneficial for the daily endeavors of fitness coaches, with repercussions for the exercisers' experiences and their outcomes from exercise.

## Conclusions

To the best of, no research has explored psychological and physiological variables in an integrated and longitudinal way in the scope of exercise in health/fitness clubs. Hence, the data gathered through this investigation have the potential to bring a novel perspective to current research in the field. According to the findings of this research, the fitness coach seems to be a central character in the individual's involvement with exercise, their outcomes, and adherence. These data arises from the evidence of their potential to have a direct impact on the exerciser's intrinsic motivation and enjoyment. In general, the categories with the highest predictive capacity for exercisers' outcomes are competence, enjoyment, and exercise planning and prescription. The results reaffirm the possibility of the fitness coaches being considered as important elements in ensuring the spread and sustainability of exercising habits, public health promotion, and the expansion of the sector.

## Acknowledgments

The authors would like to thank all participants involved.

## Author Contributions

**Conceptualization:** Ricardo Braga-Pereira, António Rodrigues Sampaio, Pedro Teques.

**Data curation:** Ricardo Braga-Pereira, Guilherme Eustáquio Furtado, Francisco Campos.

**Formal analysis:** Ricardo Braga-Pereira, Pedro Teques.

**Investigation:** Ricardo Braga-Pereira, Guilherme Eustáquio Furtado, Francisco Campos, António Rodrigues Sampaio.

**Methodology:** Ricardo Braga-Pereira, António Rodrigues Sampaio, Pedro Teques.

**Supervision:** António Rodrigues Sampaio, Pedro Teques.

**Validation:** António Rodrigues Sampaio, Pedro Teques.

**Writing – original draft:** Ricardo Braga-Pereira, Guilherme Eustáquio Furtado.

**Writing – review & editing:** Ricardo Braga-Pereira, Pedro Teques.

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
