## [Decision Letter · Decision Letter 0]

10 Jun 2024

PONE-D-24-12195Impact of fitness coach behavior on exercise motivation, commitment, and enjoyment: A longitudinal studyPLOS ONE

Dear Dr. Sampaio,

Thank you for submitting your manuscript to PLOS ONE. After careful consideration, we feel that it has merit but does not fully meet PLOS ONE’s publication criteria as it currently stands. Therefore, we invite you to submit a revised version of the manuscript that addresses the points raised during the review process.

We look forward to receiving your revised manuscript.

Kind regards,

Leonardo Vidal Andreato, PhD

Academic Editor

PLOS ONE

Journal Requirements:

**Additional Editor Comments:**

**ACADEMIC EDITOR: **Dear author, as you can see, although the reviewers recognized the merit of the study, they had doubts regarding the submitted paper. To submit a new version of this study, all questions must be answered in a response letter and changes must be highlighted in the manuscript.

Reviewers' comments:

Reviewer's Responses to Questions

**Comments to the Author**

1. Is the manuscript technically sound, and do the data support the conclusions?

Reviewer #1: No

Reviewer #2: Yes

2. Has the statistical analysis been performed appropriately and rigorously? 

Reviewer #1: No

Reviewer #2: Yes

3. Have the authors made all data underlying the findings in their manuscript fully available?

Reviewer #1: Yes

Reviewer #2: Yes

4. Is the manuscript presented in an intelligible fashion and written in standard English?

Reviewer #1: Yes

Reviewer #2: Yes

5. Review Comments to the Author

Reviewer #1: Dear authors,

The study has some flaws that undermine the quality of the research.

I highlight some points:

1. The introduction is too long, a vast review of concepts is not necessary, I suggest reducing it.

2. Some questions are in the method, but are results and others are in the result, but are part of the method. Number of participants, average age, for example, should go into the result. Statistical analysis done to determine the size of the sample method.

3. There is a lack of characterization of the participants, in addition to the fact that they are men or women, issues such as practice time, which modality practiced, frequency, would be fundamental to understanding the data. The regression between the questionnaires and the physical variables tells us nothing.

4. The results of the questionnaires should be presented in other ways and discussed. For example, what is the perception of exercisers about the trainer? Is participants’ motivation more intrinsic or extrinsic? How is it regulated? What is the level of pleasure of the participants? High or low?

Reviewer #2: I would like to thank the authors the opprtunite to read your paper. It was a pleasure to understand the idea of the project and follow the entire process until reading the findings. In fact, only one issue emerged after reading the Statistics section. Why the authors prefered to apply a t test insted an ANOVA? It was not clear to me if tha authors compared the baseline measures with the three and six months isolated or why they did not apply a one-way ANOVA.

6. PLOS authors have the option to publish the peer review history of their article (what does this mean?). If published, this will include your full peer review and any attached files.

Reviewer #1: No

Reviewer #2: **Yes: **Marcelo Marques

---

## [Author Response · Author response to Decision Letter 0]

25 Jul 2024

Responses to reviewers and the editor were attached in the corresponding field according to the suggested procedure.

---

## [Decision Letter · Decision Letter 1]

9 Sep 2024

Impact of fitness coach behavior on exercise motivation, commitment, and enjoyment: A longitudinal study

PONE-D-24-12195R1

Dear Dr. Sampaio,

We’re pleased to inform you that your manuscript has been judged scientifically suitable for publication and will be formally accepted for publication once it meets all outstanding technical requirements.

Kind regards,

Leonardo Vidal Andreato, PhD

Academic Editor

PLOS ONE

Additional Editor Comments (optional):

Reviewers' comments:

Reviewer's Responses to Questions

**Comments to the Author**

1. If the authors have adequately addressed your comments raised in a previous round of review and you feel that this manuscript is now acceptable for publication, you may indicate that here to bypass the “Comments to the Author” section, enter your conflict of interest statement in the “Confidential to Editor” section, and submit your "Accept" recommendation.

Reviewer #1: All comments have been addressed

Reviewer #2: All comments have been addressed

2. Is the manuscript technically sound, and do the data support the conclusions?

Reviewer #1: Yes

Reviewer #2: Yes

3. Has the statistical analysis been performed appropriately and rigorously? 

Reviewer #1: Yes

Reviewer #2: Yes

4. Have the authors made all data underlying the findings in their manuscript fully available?

Reviewer #1: Yes

Reviewer #2: Yes

5. Is the manuscript presented in an intelligible fashion and written in standard English?

Reviewer #1: Yes

Reviewer #2: Yes

6. Review Comments to the Author

Reviewer #1: (No Response)

Reviewer #2: I would like to thank the authors the opprtunite to read this revised version of your paper. It was a pleasure to understand the idea of the project and follow the entire process until reading the findings.

7. PLOS authors have the option to publish the peer review history of their article (what does this mean?). If published, this will include your full peer review and any attached files.

Reviewer #1: No

Reviewer #2: **Yes: **Marcelo Marques

---

## [Editor Report · Acceptance letter]

17 Sep 2024

PONE-D-24-12195R1 

PLOS ONE

Dear Dr. Sampaio, 

I'm pleased to inform you that your manuscript has been deemed suitable for publication in PLOS ONE. Congratulations! Your manuscript is now being handed over to our production team.

Kind regards, 

on behalf of

Dr. Leonardo Vidal Andreato 

Academic Editor

PLOS ONE